# Evaluating the Image Quality of Neck Structures Scanned on Chest CT with Low-Concentration-Iodine Contrast Media

Jimin Kim [ID], Jee-Young Kim *, Se-Won Oh [ID] and Hyun-Gi Kim [ID]

Department of Radiology, Eunpyeong St. Mary's Hospital, College of Medicine, The Catholic University of Korea, Seoul 03312, Republic of Korea
* Correspondence: jeeyoungkim@catholic.ac.kr; Tel.: +82-2-2030-3011; Fax: +82-2-2030-3026

**Abstract:** Background: The purpose of this study was to investigate and compare the image quality of low-concentration-iodine (240 mgI/mL) contrast media (CM) and high-concentration-iodine (320 mgI/mL) CM according to the radiation dose. Methods: A total of 366 CT examinations were examined. Based on an assessment of quantitative and qualitative parameters by two radiologists, the quality was compared between Group A (low-concentration-iodine CM) and Group B (high-concentration-iodine CM) images of thyroid gland, sternocleidomastoid muscle (SCM), internal jugular vein (IJV), and common carotid artery (CCA). Another subgroup analysis compared Group a, (using ≤90 kVp in Group A), and Group b, (using ≥100 kVp in Group B) for finding the difference in image quality when the tube voltage is lowered. Results: Image quality did not differ between Groups A and B or between Groups a and b. The signal-to-noise ratio and contrast-to-noise ratio were significantly higher for Group B than Group A for the thyroid gland, IJV, and CCA. No statistical differences were found in the comparison of all structures between Groups a and b. Conclusion: There was no significant difference in image quality based on CM concentration with variable radiation doses. Therefore, if an appropriate CT protocol is applied, clinically feasible neck CT images can be obtained even using low-concentration-iodine CM.

**Keywords:** contrast media; computed tomography; iodine concentration; radiation dose; image quality





## 1. Introduction

The use of iodinated contrast agents in CT scans improves the visibility of vascular structures and organs. Their value has long been recognized [1]. Therefore, neck computed tomography (CT) in combination with iodinated contrast agents could be a standard medical imaging technique for studying various conditions in the neck, such as inflammation, neoplasm, and trauma; in addition, it can be performed in emergency settings [2,3]. Due to its relatively short scan duration and good reproducibility, this method is widely used to identify anatomical structures, diagnose pathologies, and evaluate treatment responses [4]. However, with contrast-enhanced CT, the administration of iodine contrast media (CM) [5–7] and radiation exposure [8,9] cause unforeseen problems, which are considered as general disadvantages of CT examinations. CM-related side effects are known to occur in fewer than 1.0% of patients who undergo CT scans with contrast enhancement; however, these are more likely to occur in patients with an underlying history, such as previous allergic reactions or renal failure [1,5]. According to a previous meta-analysis by Caro et al. [10], the risk of death with highly osmotic CM was as much as 0.9 per 100,000 injections. In addition, contrast-induced nephropathy (CIN) is one of the most serious, though rare, complications related to CM administration and can result in irreversible kidney failure [11–13]. Moreover, patients with pre-existing thyroid disease are at risk of iodine-induced hypothyroidism or hyperthyroidism after iodine challenge [14]. Decreased intrathyroidal deiodinase activity due to increased iodine load may also contribute to a decrease in thyroid hormone synthesis [15]. Thus, low-concentration-iodine CM is recommended to avoid these adverse events [16–18].

Radiation exposure is another concern when considering CT scans. Due to its nature, CT involves a higher radiation dose than typical conventional X-ray imaging [8]. It is also estimated that more than 62 million CT scans are currently performed annually in the United States, including on at least 4 million children [19]. It is crucial to perform CT scans with the lowest possible radiation dose for patients who need to undergo repeated CT examinations or who are relatively vulnerable to radiation exposure, such as pediatric patients [8,9]. Hence, many studies have attempted to reduce the radiation dose by using a low tube voltage while maintaining image quality [20–23].

The relative attenuation of iodinated contrast material is increased at a lower kVp because the X-ray absorption of iodine increases substantially with lower effective beam energies as long as the effective energy is maintained above the iodine angle (33 keV) [24]. Current approaches to reduce radiation exposure during CT examinations utilize automatic dose modulation strategies, typically based on low tube voltage coupled with iterative reconstruction and other dose reduction techniques [25]. Also, in addition, several papers have reported the diagnostic potential of low kVp for reducing iodinated contrast agent load in CT [21,26–28]. Of course, radiologists may be reluctant to use low-concentration-iodine CM because the reduced iodine load may affect diagnostic performance. A previous study showed that with low-dose—but not low-concentration—iodine CM, neck CT had a sufficient diagnostic potential for the assessment of preoperative thyroid cancer when compared to high-dose iodine CM. However, the combination compared in this study included a significantly lower tube voltage as well as a lower dose of contrast agent [22]. Low-voltage CT has also been studied with regard to neck CT for evaluating cervical soft tissue and bone with remarkable results [27]. Therefore, we sought to discern the applicability of this technique to a wider variety of cervical structures than that investigated in the previous study, not only for the difference in tube voltage, but also for the difference in iodine concentration.

This study aimed to evaluate the clinical applicability of low-concentration- iodine CM for the neck organs by conducting quantitative and qualitative image analysis using CM with different iodine concentrations according to various tube voltages.

## 2. Material and Methods

The institutional review board of our hospital approved this study and waived the requirement for informed consent due to its retrospective design.

### 2.1. Study Population

Initially, 436 patients who were scheduled to undergo chest CT including the thyroid in our institution between April 2019 and October 2020 were reviewed. We did not use 240 mgI/mL CM directly for neck CT because it has not yet been demonstrated as clinically applicable for neck CT. However, 240 mgI/mL CM had previously been used for chest CT in our institution as the lung and mediastinum images presented obvious contrast [28]. Therefore, we evaluated the neck structures included in chest CT. We excluded 70 patients who underwent total thyroidectomy and patients with severe artifacts that did not meet the inclusion criteria. Figure 1 summarizes the inclusion and exclusion criteria and presents a flowchart of the overall enrollment process.

Eventually, 366 patients ranging from 29 to 95 years (mean ± standard deviation (SD) age: 66.4 ± 12.3 years) in age were enrolled in this study. The patients were randomly administered CM with different iodine concentrations (240 mgI/mL iohexol (Iobrix 240, TAEJOON pharm Co, Seoul, Republic of Korea) and 320 mgI/mL ioversol (Optiray 320, Villepinte, France)). We classified patients into two different groups according to the administered iodine CM concentration: Group A with 240 mgI/mL and Group B with 320 mgI/mL. We also carried out a comparison between Group a, which consisted of Group A patients who underwent CT with a lower tube voltage (≤90 kVp), and Group b, which consisted of Group B patients who underwent CT with a higher tube voltage (≥100 kVp), to assess the difference in image quality when the tube voltage was lowered.

A total of 366 CT examinations (Group A: 159 examinations; Group B: 207 examinations; Group a: 51 examinations; Group b: 143 examinations) were included in this study. Clinical information such as age, sex, body weight, and height were collected from electronic medical records. Information related to radiation dose, including tube voltage, tube current, volume CT dose index (CTDIvol), and dose length product (DLP), was collected from the CT dose reports. The volume of CM used in each examination was recorded during the procedure. The collected information was used to calculate the amount of iodine in the CM.

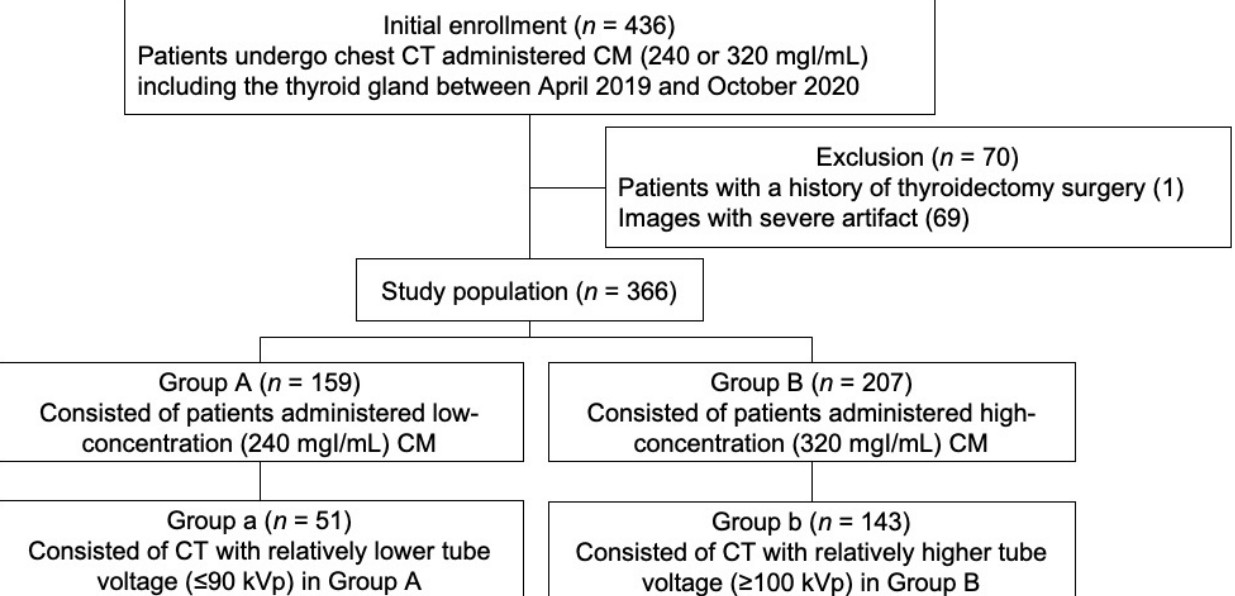

**Figure 1.** Flowchart of the enrollment of the study population, CT = computed tomography, CM = contrast media.

*2.2. CT Protocol*

The CT examinations were performed using two different CT machines. Machine A was a 128-slice single-source CT scanner (SomatomEdge, Siemens Healthineers, Erlangen, Germany) with a usable tube potential of 70–140 kVp, 20–800 mA; machine B was a dual-source 384-slice (2 × 192) CT (SomatomForce, Siemens Healthineers). Variable tube potential (70–150 kVp and 20–1300 mA) CT images were taken 55 s after CM injection (1.4 mL/kg, 2 mL/s) with 20 mL saline flushing via the antecubital vein using a power injector (Medrad injector, Medrad, Warrendale, PA, USA). The acquisition parameters were similar on both two machines: slice thickness, 3 mm at 3 mm intervals; rotation time, 0.5 s; pitch, 1; automatic tube voltage modulation (CARE kV, Siemens Healthineers) using reference kV 120; automatic tube current selection (CAREDose 4D, Siemens Healthineers) using reference mAs 130; collimation 128 × 0.6 for machine A and 192 × 0.6 for machine B.

*2.3. Image Analyses*

Postcontrast images were evaluated using a ZeTTA PACS workstation (TaeYoung Soft, Seoul, Republic of Korea). For qualitative analyses, two independent radiologists who were blinded to the protocol evaluated the quality of the CT images that were reconstructed with soft kernel (Br40). The readers could readjust the window width and level. Image sharpness, noise, and comprehensive image quality were determined on 3-, 3-, and 5-point scales, respectively, referring to previous studies (Table 1) [21,22]. Before proceeding with the evaluation, the two radiologists set a reference picture for scores and set the eye level. The mean scores provided by the two radiologists were used for statistical analysis.

**Table 1.** Qualitative scales of image sharpness, noise, and overall image quality.

| Score | Sharpness | Artifact | Overall Image Quality |
|---|---|---|---|
| 1 | Blurred contour | Interfering structure | Poor |
| 2 | Average | No interfering structure | Suboptimal |
| 3 | Sharp contour | No artifact | Acceptable |
| 4 | | | Good |
| 5 | | | Excellent |

Quantitative image analysis was performed by two trained researchers using the PACS system in our institution. They identified five regions of interest (ROIs) to assess the Hounsfield unit (HU) and SD of the thyroid gland, sternocleidomastoid muscle (SCM), internal jugular vein (IJV), common carotid artery (CCA), and subcutaneous fat of the posterior neck with circular ROIs manually drawn as large as possible (Figure 2). These measurements were reviewed by one radiologist for appropriateness.

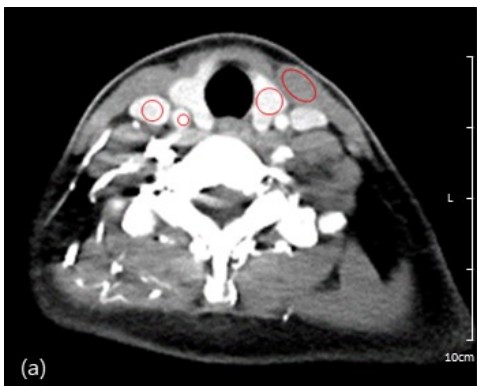 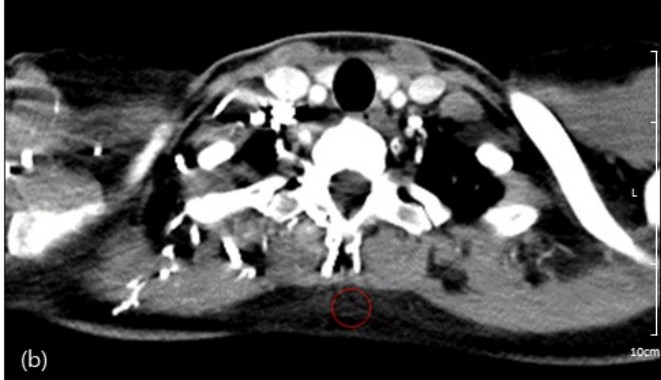

**Figure 2.** (**a**) Quantitative analysis of CT images on which two trained researchers drew five regions of interest in the thyroid gland, sternocleidomastoid muscle (SCM), internal jugular vein (IJV), common carotid artery (CCA); and (**b**) subcutaneous fat of the posterior neck.

The signal-to-noise ratio (SNR) and contrast-to-noise ratio (CNR) were calculated in the thyroid gland, SCM, IJV, and CCA as follows:

$$\text{SNR} = \text{mean attenuation value}/\text{image noise} \tag{1}$$

Image noise was defined as the SD of the attenuation measured in the subcutaneous fat for the thyroid gland, SCM, IJV, and CCA. The CNR was calculated as follows:

$$\text{CNR} = (\text{mean attenuation value} - \text{mean attenuation of reference tissue})/\text{image noise} \tag{2}$$

Reference tissue and image noise were determined to calculate the mean attenuation and SD of subcutaneous fat, respectively.

### 2.4. Radiation Dose

The CTDIvol and DLP provided by each CT scanner workstation were saved as Digital Imaging and Communications in Medicine files. The effective dose is directly proportional to the total radiation dose scanned and the overall risk of the irradiated tissue [29]. Therefore, we converted DLP to normalized effective dose using the conversion factors (0.017) reported in publication 103 of the International Commission on Radiological Protection 3 and the European guidelines for multislice computed tomography [30,31].

$$\text{Effective dose (mSv)} = \text{conversion coefficient } [\text{mSv}/(\text{mGy} \times \text{cm})] \times \text{DLP (mGy} \times \text{cm)} \tag{3}$$

*2.5. Statistical Analysis*

All statistical analyses were performed using SPSS (version 21.0, IBM Corp., Armonk, NY, USA) and all graphs were created using GraphPad Prism (version 8.4.2, GraphPad Software Inc., San Diego, CA, USA). The comparison of patient characteristics, radiation exposure, and quantitative and qualitative parameters between groups was based on a Student's *t*-test. Image quality was also compared between the two CT machines using a Student's *t*-test.

The interobserver agreements between the two radiologists for the quantitative and qualitative analysis results were evaluated based on kappa values. The kappa values were interpreted as follows: less than 0.40 denotes poor agreement, 0.40 to less than 0.60 denotes fair agreement, 0.60 to less than 0.80 denotes good agreement, and 0.80 to 1.00 denotes excellent agreement. $p < 0.05$ was considered to indicate a statistically significant difference.

## 3. Results

*3.1. Demographics*

The patient demographics are summarized in Table 2. There were no significant differences in patient demographics and the CT examinations between Group A and B and between Group a and Group b. Iodine amount was significantly higher in Group A than in Group B, while tube voltage and iodine amount were significantly higher in Group a than Group b. In addition, since the same amount of CM was used for both groups, the amount of iodine per kg (i.e., total iodine load) in groups A and B was 571 and 363 mg/kg, respectively. Body mass index was calculated as follows:

$$\text{Body mass index } \left(\text{kg/m}^2\right) = \text{Weight (kg)}/[\text{Height (m)}]^2$$

**Table 2.** Mean value of patient characteristics.

|  | Group A($n$ = 159) | Group B($n$ = 207) | $p$-Value | Group a ($n$ = 51) | Group b($n$ = 143) | $p$-Value |
|---|---|---|---|---|---|---|
| Age (y) | 66.96 | 66.16 | 0.688 | 67.24 | 65.66 | 0.358 |
| Sex (M:F) | 92:67 | 105:102 | 0.175 | 21:30 | 88:55 | 0.204 |
| Weight (kg) | 60.00 | 59.19 | 0.348 | 54.49 | 61.94 | 0.914 |
| Height (cm) | 161.31 | 160.44 | 0.577 | 157.61 | 162.32 | 0.873 |
| Body mass index(kg/m$^2$) | 22.85 | 22.70 | 0.169 | 22.01 | 23.26 | 0.433 |
| Tube voltage (kVp) | 97.67 | 97.53 | 0.959 | 88.04 | 102.45 | <0.001 * |
| CT machine (machine A:B) | 77:82 | 107:100 | 0.537 | 17:34 | 78:65 | 0.917 |
| Iodine amount (g) | 21.57 | 29.42 | <0.001 * | 24.77 | 30.21 | <0.001 * |

* = $p$-value < 0.05, Student's *t*-test.

*3.2. Qualitative Image Analyses*

The subjective qualitative analysis results are summarized in Table 3. When comparing Group A with Group B and Group a with Group b, there were no significant differences in any of the qualitative scales; these groups showed acceptable overall image quality (Figure 3). However, images from the group that underwent CT in machine B presented significantly higher performance than those from machine A. In the subanalysis comparing machine A and machine B in terms of differences in contrast agent concentration (Tables S1 and S3) in combination with tube voltage (Tables S2 and S4), there was no difference in image quality between the two machines. The interobserver agreement of each variable was fair to good (Table 4).

**Table 3.** Results of qualitative image analysis according to parameters.

| | Group A | Group B | *p*-Value | Group a | Group b | *p*-Value | Machine A | Machine B | *p*-Value |
|---|---|---|---|---|---|---|---|---|---|
| Sharpness | 2.91 | 2.96 | 0.052 | 2.93 | 2.96 | 0.214 | 2.93 | 2.94 | 0.297 |
| Noise | 2.26 | 2.26 | 0.916 | 2.30 | 2.23 | 0.340 | 2.20 | 2.33 | 0.006 * |
| Overall image quality | 3.21 | 3.22 | 0.885 | 3.25 | 3.20 | 0.428 | 3.16 | 3.28 | 0.014 * |

\* = *p*-value < 0.05, Student's *t*-test.

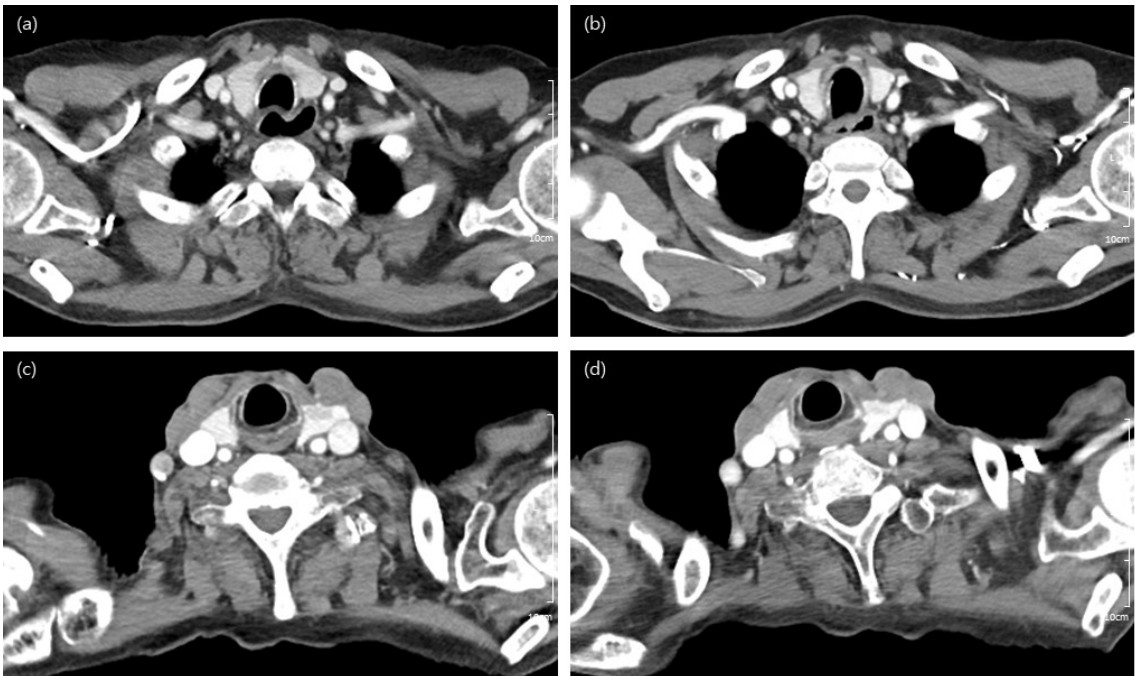

**Figure 3.** Computed tomography (CT) images of a 60-year-old woman obtained using low-concentration-iodine (240 mgI/mL) CM (contrast media), corresponding to Group A (**a**); follow-up image using high-concentration-iodine (320 mgI/mL) CM, corresponding to Group B (**b**). Both images have same tube voltage (100 kVp). Although there is a little streak artifact in the Group A image, both radiologists considered both images to be of good to excellent overall image quality. CT images of a 73-year-old man obtained using low-concentration-iodine (240 mgI/mL) CM and low tube voltage (90 kVp), corresponding to Group a (**c**); and follow-up image using high-concentration-iodine (320 mgI/mL) CM, corresponding to Group b (**d**). Despite the streak artifact present in the Group a image, both images were considered to exhibit good overall image quality.

**Table 4.** Interobserver agreement for each variable.

| | *k* * |
|---|---|
| Sharpness | 0.658 |
| Noise | 0.586 |
| Overall image quality | 0.718 |

\* = Kappa value of overall interobserver agreement.

### 3.3. Quantitative Image Analyses

The quantitative analysis results are summarized in Table 5. The mean attenuation value was significantly higher for Group B than Group A for the SCM, IJV, CCA, and thyroid gland but not for subcutaneous fat. When Group a and Group b were compared, there was no statistical difference in attenuation, CNR, and SNR for all estimated structures (Figure 4).

**Table 5.** The mean attenuation, SNR, and CNR of quantitative analysis.

|  |  | Group A | Group B | *p*-Value | Group a | Group b | *p*-Value |
|---|---|---|---|---|---|---|---|
| | Fat | −93.58 | −94.23 | 0.427 | −96.73 | −93.30 | 0.216 |
| | SCM | 73.59 | 76.16 | 0.028 * | 70.10 | 75.47 | 0.324 |
| Attenuation | IJV | 204.01 | 252.98 | <0.001 * | 217.53 | 236.60 | 0.439 |
| | CCA | 234.14 | 312.49 | <0.001 * | 253.36 | 297.90 | 0.185 |
| | Thyroid | 188.67 | 220.31 | <0.001 * | 194.49 | 214.45 | 0.087 |
| | SCM | 6.34 | 6.21 | 0.467 | 6.09 | 6.14 | 0.492 |
| SNR | IJV | 17.74 | 20.58 | <0.001 * | 19.07 | 19.27 | 0.788 |
| | CCA | 20.29 | 25.73 | <0.001 * | 22.10 | 24.45 | 0.509 |
| | Thyroid | 16.28 | 17.99 | 0.008 * | 16.84 | 17.52 | 0.384 |
| | SCM | 14.42 | 13.99 | 0.256 | 14.53 | 13.84 | 0.592 |
| CNR | IJV | 25.81 | 28.36 | 0.024 * | 27.50 | 26.97 | 0.867 |
| | CCA | 28.37 | 33.51 | <0.001 * | 30.54 | 32.16 | 0.616 |
| | Thyroid | 24.36 | 25.77 | 0.167 | 70.10 | 75.47 | 0.570 |

SNR = signal-to-noise ratio, CNR = contrast-to-noise ratio, * = *p*-value < 0.05, Student's *t*-test.

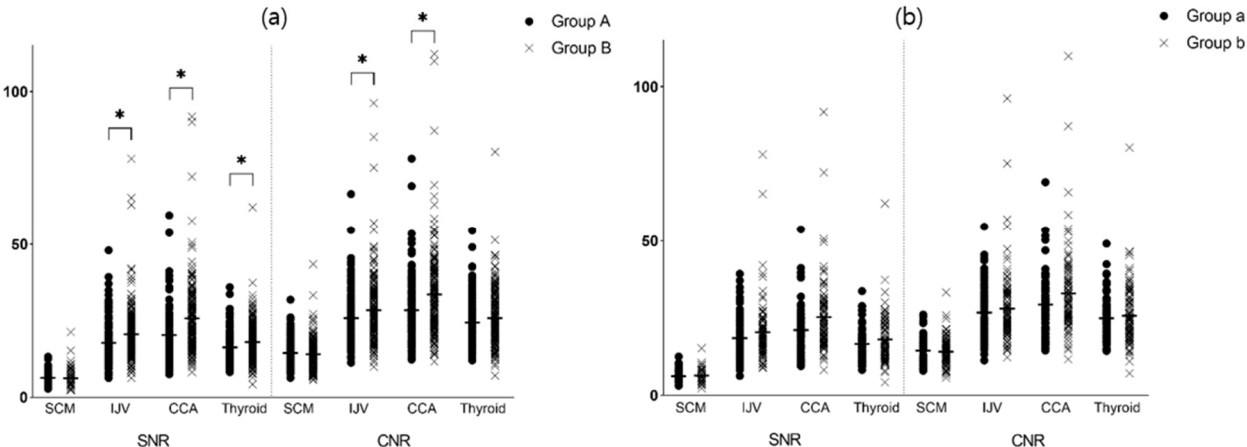

**Figure 4.** Differences in SNR and CNR between (**a**) Group A and B; and (**b**) Group a and b. * = *p*-value < 0.05, Student's *t*-test.

### 3.4. Estimation of Radiation Dose

The CTDIvol, DLP, and effective doses are summarized in Table 6. Radiation exposure was significantly lower in Group a than Group b (*p*-value < 0.05, all variables), with a 19.9% reduction in the effective dose. The difference in effective dose between Groups A and B was not significant (Table S5).

**Table 6.** The CTDIvol, DLP, and effective doses of Group a and Group b.

|  | Group a | Group b | *p*-Value |
|---|---|---|---|
| CTDIvol (mGy) | 5.39 ± 1.58 | 6.14 ± 2.13 | 0.020 * |
| DLP (mGy × cm) | 195.66 ± 56.59 | 244.18 ± 90.67 | <0.001 * |
| Effective dose (mSv) | 2.93 ± 0.85 | 3.67 ± 1.36 | <0.001 * |
| Dose reduction (%) | −19.9 | | |

CTDIvol = volume CT dose index, DLP = dose length product. * = *p*-value < 0.05, Student's *t*-test.

### 4. Discussion

We evaluated the applicability of low-concentration-iodine (240 mgI/mL) CM for neck structures imaged using chest CT. Compared to previous research, not only soft tissues but also more thyroid and vascular neck structures from a much larger population were included in this study [27]. We also compared groups according to differences in iodine

concentration and tube voltage. In the qualitative image assessment, there was no statistical difference between the overall image qualities according to iodine amount or tube voltage. In the quantitative image assessment, the HU and SNR of vessels (IJV and CCA) and thyroid glands were significantly higher in Group B than Group A. The attenuation of vessels and thyroid glands seemed to be affected more by CM concentration, likely because they have a relatively higher vascularity than SCM or subcutaneous fat. The HU, SNR, and CNR of SCM and subcutaneous fat did not differ between Group A and Group B. In addition, there was no significant difference between SNR and CNR between Group a and Group b.

The qualitative image quality of neck structure can be maintained for neck structures when low-concentration-iodine CM is used with moderate to good interobserver agreement, which could be significant enough according to previous studies [22,23,29]. In particular, both the qualitative and quantitative image quality was maintained while the effective dose was reduced in contrast-enhanced CT with a lower radiation tube voltage using low-concentration-iodine CM. With respect to the average dose of 7 mSv listed by the US Food and Drug Administration, our hospital showed significantly lower values in both groups [30]. Over the past decade, several papers have reported the potential for reducing iodinated contrast agent load in CT, especially in combination with low-kVp scanning [24,27,32,33]. This is possible because the role of iodine is fundamental in low-dose CT. Low X-ray energies approach the iodine k-absorption edge (33 keV), which increases the HU value of CT [25]. The high iodine attenuation obtained at low kVp settings can reduce radiation exposure while maintaining SNR [26,34]. Advanced reconstruction techniques, such as low tube voltage and iterative reconstruction, are usually used for the weak enhancement of low-concentration-iodine CM and a reduction in iodine loading.

In this study, we also analyzed the difference in image quality between the two CT machines. Machine B produced images with less noise and therefore higher image quality than machine A. In general, dual-source CT shows superior performance to over single-source CT due to a wider range of tube voltage and tube current [31,32]. This is because data acquisition with two different voltages produces large spectral separation using additional filtering and consequently material separation without affecting the dose [33–35]. This allow differences in CT equipment to affect image quality. Automatic tube voltage selection and automatic tube current modulation can both affect image quality; however, in this study, both techniques were applied on both machines with similar settings. Therefore, these factors would not have influenced our study results.

This study has several limitations. First, we used the protocol for chest CT in this study. In doing so, the exact nature of the neck may not have been adequately reflected because certain factors, such as CM time delay after injection, usually differ between contrast-enhanced chest and neck CT protocols. However, since the images were reconstructed using the same soft tissue kernel that is used for the neck CT, the texture of the CT images was similar to that of neck CT images. Second, due to the retrospective design of this study, selection bias cannot be excluded. Moreover, the boundary values of tube voltage are not strict in the same context because, unlike in our study, many prospective studies have used voltages as low as 70 or 80 kVp for CT [21,22,27,36]. These are inevitable limitations, as the findings of this study are based on retrospective research. Finally, we compared the normal structures of the neck; this was slightly different from evaluating the structure of the pathologic state in CT using a low concentration of CM. Further work is required to determine the optimal protocol that balances radiation dose and iodine concentration with the maintenance of suitable quality in the diagnostic imaging of pathological conditions.

## 5. Conclusions

In conclusion, there was no significant difference in the qualitative image quality of CT scans for the clinical evaluation of neck organs by radiologists due to the iodine concentration of the CM. Further prospective studies exploring neck CT protocols based on a combination of different iodine concentrations for variable states of the neck will be

necessary to determine the generalizability of our findings. Therefore, given the application of an appropriate CT protocol, clinically feasible neck CT images can be obtained using low-concentration-iodine CM.

**Supplementary Materials:** The following supporting information can be downloaded at: https://www.mdpi.com/article/10.3390/tomography8060239/s1, Table S1: In machine A, comparing the group with 240 iodine contrast media (CM) and the group with 320 iodine CM; Table S2: In machine A, comparing the group with 240 iodine CM and relatively low kVp (90≥) and the group with 320 CM media and relatively high kVp (100≤); Table S3: In machine B, comparing the group with 240 iodine CM and the group with 320 iodine CM; Table S4: In machine B, comparing the group with 240 iodine CM and relatively low kVp (90≥) and the group with 320 iodine CM and relatively high kVp (100≤); Table S5: Effective doses between Group A and Group B.

**Author Contributions:** Conceptualization, J.-Y.K. and S.-W.O.; Data curation, J.K. and J.-Y.K.; Formal analysis, J.K. and J.-Y.K.; Funding acquisition, J.-Y.K. and S.-W.O.; Investigation, J.K. and J.-Y.K.; Methodology, J.K. and J.-Y.K.; Project administration, J.-Y.K. and S.-W.O.; Supervision, J.-Y.K.; Validation, J.K. and J.-Y.K.; Visualization, J.K.; Writing—original draft, J.K.; Writing—review and editing, J.K., J-Y.K., S.-W.O. and H.-G.K. All authors have read and agreed to the published version of the manuscript.

**Funding:** This study was funded by a Research Fund from TAEJOON pharm Co., Ltd. Republic of Korea. The funders had no role in study design, data collection and analysis, decision to publish, or preparation of the manuscript.

**Institutional Review Board Statement:** This study was performed in line with the principles of the Declaration of Helsinki. Approval was granted by the Institutional Review Board of Eunpyeong St. Mary's Hospital, the Catholic University of Korea (Protocol number PC20EISI0093, approval date 2 July 2020), retrospectively designed.

**Informed Consent Statement:** The institutional review board of our hospital has waived the requirement for informed consent due to its retrospective study design.

**Data Availability Statement:** The data presented in this study are available on request from the corresponding author. The data are not publicly available due to privacy or ethical concerns.

**Acknowledgments:** This work was supported by the Catholic University of Korea. The authors thank Christine Y. Park for the word processing.

**Conflicts of Interest:** The authors have no relevant financial or non-financial interest to disclose.

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
