# Peer review of "Evaluating the Image Quality of Neck Structures Scanned on Chest CT with Low-Concentration-Iodine Contrast Media"

_tomography, doi:10.3390/tomography8060239_

Round 1

Reviewer 1 Report (New Reviewer)

Kim et al evaluated Neck CT in 366 patients with different iodine concentrations. They concluded that low iodine and high iodine dosages produce similar quality images. Although the study has high merit, it has several flaws which must be corrected. I have mentioned them below.

(1) Please explain what does the values in Table 2 signifiy? Do they mean mean or median numbers of the demographics. For example, the weight of 60 kgs and 59.19 kgs in groups A and B are too close to each other and seems like the authors found very similar patients. 

(2) Line 186 states that The subjective qualitative analysis results are summarized in Table 3. The sharpness 186 score was significantly higher in Group B (2.96 ± 0.18) than A (2.91 ± 0.24). The difference of 0.05 between 2.91 and 2.96 are smaller than the errors of .18 and .24. This statement of significantly higher score for Group B doesn't hold true. 

(3) The authors should explain the safe radiation levels. 

(4) In figure 1, it is mentioned that 51 patients in group A with lower iodine dose get lower radiation and 143 patents in group b with higher iodine get higher dose. Did the rest of the patients get equal dose? Please include a table mentioning these results clearly with different dose levels.

(5) The images dont have scale bars. Please put scale bars.

Author Response

Reviewer 2 Report (New Reviewer)

1. Neck CT was used in the title and introduction, but the study data in the methods was chest CT. The reviewer was confused about why chest CT was used rather than neck CT. Although the reason for using chest CT was mentioned in the study population section (Lines 87-91), the introduction kept telling is neck CT. The author should revise the introduction section to focus on neck CT.

2. The statement “but not low-iodine-concentration, CM has diagnostic potential...” is not accurate (Lines 72-73). What Ref 21 has studied includes low Iodine concentration. They evaluated the image quality of 70-kVp thyroid CT with low volumes of CM. Thus, I would like the author to comment in detail on the differences between this paper and the previous study.

3. Table 4 shows that the agreement between observers is not good enough. How was each variable calculated when large difference between the two observers?

4. The CT images were acquired with machines A and B. Line 90 states that the imaging quality of machine B was significantly better than A. The authors did not exclude the effect of the imaging machines on the study results. For groups A and B, groups a and b, if the patients are further divided into subgroups according to imaging machines, do the study results still support the paper’s conclusions?

5. What is the basis for the evaluation of the scores in Table 1 ? Did the observer's subjective assessment?

6. In Tables 3 and 5, the comparison results of groups A and B, groups a and b showed significant differences. Could the authors address the possible reasons? And how did the authors conclude “Image quality did not differ between Groups A and B or between Groups a and b” (Line 24)?

7. In the title, should low-concentration-iodine be low-concentration-iodine? As elsewhere in the paper was written as low-concentration-iodine.

8. The statement “the more …, the more…” is too colloquial (Lines 39-42).

Author Response

Reviewer 3 Report (New Reviewer)

The manuscript from Kim et al presents an interesting report about the image quality of neck CT with low and high-concentration iodine contrast media. The report addresses clinically relevant problems with results based on experimental evidence. The manuscript should be benefited from the following comments:

1.    Introduction should be more elaborate. For instance, why Iodine is used in CT scans should be introduced first to give the reader a basic understanding of the problem

2.    All the images should have a scale bar.

3.    Line 189-191: authors mentioned machines A & B and their performance. However, it is not simple to understand how that influence the results.

4.    More details about the statistical test performed are required and it would be great if provided in each figure/table legend.

Round 2

Reviewer 1 Report (New Reviewer)

agreed

Reviewer 2 Report (New Reviewer)

Point 7: Low--iodine-concentration is not unified to "low-concentration-iodine" in the abstract and somewhere in the text.  Please check it thoroughly again.

This manuscript is a resubmission of an earlier submission. The following is a list of the peer review reports and author responses from that submission.

Round 1

Reviewer 1 Report

This paper evaluates quality in contrast-enhanced neck CT, when performed with two concentrations of contrast media. This is performed in a retrospective study on a patient collective that is scanned for chest CT (not neck). 

Last decade, several papers reported the possibilities of reducing iodine contrast agent load in CT, especially in combination with low kVp scanning. This is possible because low X-ray energies approach the iodine k-absorption edge (34 keV) which increases the CT signal (HU-values). Hence, the iodine load can be reduced. Unfortunately, this paper does not add lot to the current knowledge. In addition, several flaws can be identified in the study design which prevent publication.

Major comments:

- In the introduction section the authors refer to publications on contrast safety which are +30 years old. I advise to take a look at the 2021 ACR Manual on Contrast Media from the commission on Quality and Safety as a guide for radiologists to enhance the safe and effective use of contrast media

- The authors did not consider the Iodine load (mg I) between the groups, which will depend on both the concentration and the injected volume. Obviously the iodine load will have a major impact on the enhancement. This is a major shortcoming of the study

- The description of the CT protocols is very unclear and contains mistakes. For example: tube current modulation regulates mA, not kV. Scanner A seems to use two tube potentials 70-140 kVp, so are these DECT acquisitions?

- Image quality assessment. The Figure of Merit is flawed (Formula 3). This is unacceptable as it is the major IQ metric they assume. You can not normalize by effective dose! At the best you can normalize by CTDIvol. The effective dose depends on the scan length and the noise in the images does not. The results will be wrong.

In addition, how is effective dose calculated? Which methods/software models were used? The authors state ‘according ICRP-103’ but the ICRP just recommends the tissue weighting factors, not how you estimate effective dose from a CT scan.

- The quality of the text does not meet scientific standards for a medical journal. This includes inappropriate grammar, use of references, etc.

Reviewer 2 Report

In this manuscript, the authors have qualitatively and quantitatively compared normal neck structures using different chest CT protocols with different contrast media concentrations and radiation does. They showed that using lower contrast media and radiation dose resulted in no significant difference in quantitative and qualitative assessment of neck structures.

The strength of the article relies on timely addressing of an ongoing issue given the global shortage in contrast media supply. However, there are inherent limitations to the study.

- The authors are using chest CT to evaluate neck structures!

- Many institutes are using neck CT protocols with lower overall dose of contrast media and radiation protocol.

- the neck structures assessed are normal. A more relevant study would evaluate pathological structures using different imaging protocols - for example the conspicuity of oropharyngeal cancer borders using different contrast media concentration. 

Reviewer 3 Report

The authors describe a retrospective comparison between CT studied of the neck with different contrast media concentration and differente tube voltage. The aim could be of interest but there are major limitations of the study. Specific comments: I'm not sure that 320mgI/ml can be considered high concentration, since I would consider high concentration 350 or higher. Introduction: Reference 22 is too old to be considered as the background that leads the hypothesis. English must be improved. Methods: the first paragraph introduces a critical limitstion/selection bias. Chest CT were considered for neck study in the low CM concentration group. Tube voltage is automatically selected by the machine and not randomly assigned. Results: I'm not sure that the difference in average tube voltage between group a and b could be considered so relevant to be appreciated by a qualitative or simple quantitative assessment (88 kVp vs 102kVp). Conclusion: given the scientific weakness the conclusions are not supported
